# Memantine in the Prevention of Radiation-Induced Brain Damage: A Narrative Review

**DOI:** 10.3390/cancers14112736

**Published:** 2022-05-31

**Authors:** Claudia Scampoli, Silvia Cammelli, Erika Galietta, Giambattista Siepe, Milly Buwenge, Gabriella Macchia, Francesco Deodato, Savino Cilla, Lidia Strigari, Silvia Chiesa, Alessio Giuseppe Morganti

**Affiliations:** 1Department of Experimental, Diagnostic and Specialty Medicine-DIMES, Alma Mater Studiorum University of Bologna, 40138 Bologna, Italy; claudia.scampoli@studio.unibo.it (C.S.); silvia.cammelli2@unibo.it (S.C.); milly.buwenge2@unibo.it (M.B.); alessio.morganti2@unibo.it (A.G.M.); 2Radiation Oncology, IRCCS Azienda Ospedaliero, Universitaria di Bologna, 40138 Bologna, Italy; giambattista.siepe@aosp.bo.it; 3Radiation Oncology, Gemelli Molise Hospital, Università Cattolica del Sacro Cuore, 86100 Campobasso, Italy; gabriella.macchia@unicatt.it (G.M.); francesco.deodato@gemellimolise.it (F.D.); 4Medical Physics, Gemelli Molise Hospital, Università Cattolica del Sacro Cuore, 86100 Campobasso, Italy; savino.cilla@unicatt.it; 5Medical Physics, IRCCS Azienda Ospedaliero, Universitaria di Bologna, 40138 Bologna, Italy; lidia.strigari@aosp.bo.it; 6UOC di Radioterapia Oncologica, Dipartimento Diagnostica per Immagini, Radioterapia Oncologica ed Ematologia, Fondazione Policlinico Universitario Agostino Gemelli IRCCS, 00168 Rome, Italy; silvia.chiesa@policlinicogemelli.it

**Keywords:** literature review, radiotherapy, prophylaxis, memantine, brain damage

## Abstract

**Simple Summary:**

Decline in cognitive function is a major problem for patients undergoing whole-brain radiotherapy (WBRT). Scientific interest has increased due to the high dropout rate of patients in the first months after WBRT and the early onset of cognitive decline. Therefore, the study of antiglutamatergic pharmacological prophylaxis and hippocampal-sparing WBRT techniques has been deepened based on the knowledge of the mechanisms of hyperglutamatergic neurotoxicity and the role of some hippocampal areas in cognitive decline. In order to provide a summary of the evidence in this field, and to foster future research in this setting, this literature review presents current evidence on the prevention of radiation-induced cognitive decline and particularly on the role of memantine.

**Abstract:**

Preserving cognitive functions is a priority for most patients with brain metastases. Knowing the mechanisms of hyperglutamatergic neurotoxicity and the role of some hippocampal areas in cognitive decline (CD) led to testing both the antiglutamatergic pharmacological prophylaxis and hippocampal-sparing whole-brain radiotherapy (WBRT) techniques. These studies showed a relative reduction in CD four to six months after WBRT. However, the failure to achieve statistical significance in one study that tested memantine alone (RTOG 0614) led to widespread skepticism about this drug in the WBRT setting. Moreover, interest grew in the reasons for the strong patient dropout rates in the first few months after WBRT and for early CD onset. In fact, the latter can only partially be explained by subclinical tumor progression. An emerging interpretation of the (not only) cognitive impairment during and immediately after WBRT is the dysfunction of the limbic and hypothalamic system with its immune and hormonal consequences. This new understanding of WBRT-induced toxicity may represent the basis for further innovative trials. These studies should aim to: (i) evaluate in greater detail the cognitive effects and, more generally, the quality of life impairment during and immediately after WBRT; (ii) study the mechanisms producing these early effects; (iii) test in clinical studies, the modern and advanced WBRT techniques based on both hippocampal-sparing and hypothalamic-pituitary-sparing, currently evaluated only in planning studies; (iv) test new timings of antiglutamatergic drugs administration aimed at preventing not only late toxicity but also acute effects.

## 1. Introduction

Brain metastases (BMs) affect up to 30% of solid tumor patients. Whole-brain radiotherapy (WBRT) has been one of the standard treatments in patients with multiple BMs, since its introduction in the 1950s [1]. WBRT has been used to improve neurological symptoms and to reduce the risk of neurological death and intracranial progression [2,3]. However, alongside these benefits, significant treatment-related toxicity can be recorded, especially in terms of late cognitive decline (CD), with severe impairment of personal and social qualities of life [4].

The severity of these adverse, progressive, and irreversible events was more evident after the improvement in survival rates achieved in recent years thanks to advances in systemic therapies. Although stereotactic radiotherapy and molecular therapies are now the standard of care in oligometastatic disease, WBRT is still used in multiple BMs not amenable to other treatments. Therefore, the availability of strategies to prevent or treat iatrogenic CD is urgent [5].

To date, the available approaches are based on the reduced irradiation of specific critical brain areas dedicated to memory and thinking [6], and on the use of an antiglutamatergic drug, memantine, with proven anti-inflammatory properties [7]. However, clinical trials on interventions to help manage CD have had conflicting results and no standard of care has yet been established.

Therefore, in order to provide a summary of the evidence in this field, and to foster future research in this setting, the aim of this literature review is to present current evidence on the prevention of radiation-induced CD and particularly on the role of memantine.

## 2. Materials and Methods

A PubMed-based search was conducted from the first date to 25 May 2021. Only papers written in English were included. We used various combinations of the following terms, such as memantine, hippocampal-sparing, radiotherapy, neuroprotection, cognition and neuropsychology.

Abstracts of conference proceedings, study protocols, case reports, systematic or narrative reviews, meta-analyses, letter-commentaries-editorials, planning studies, imaging studies, surveys, guidelines-recommendations, or studies reporting duplicate data were excluded.

A total of 594 studies were identified. After the removal of duplicates, a first selection was made based on the titles and abstracts. Moreover, a further search in the reference list of the selected studies was conducted. Finally, 38 papers on memantine for radiation protection in patients undergoing brain radiation therapy were included in this narrative review. The narrative review checklist is shown in Appendix A.

## 3. Results

### 3.1. Memantine and Radiation-Induced Effects on the Central Nervous System

#### 3.1.1. Neuronal Degeneration and Memantine

The pathogenetic basis of neuronal degeneration relies on excessive stimulation by glutamate, the main excitatory amino acid transmitter in the central nervous system [8]. Glutamate activates neuronal receptors and starts excitatory intracellular signals. The glutamate receptors are divided into ionotropic and metabotropic [9]. Among the first, N-methyl-D-aspartate (NMDA) receptors are expressed in a broad area of the central nervous system on both neuronal and glial cells [10,11,12]. At physiological concentrations, glutamate is a fundamental regulator of neuronal plasticity [13] and is involved in learning and memory processes [14,15], as shown by studies on NMDA-dependent long-term-potentiation (LTP), neuroembryogenesis and neuronal migration [16]. However, excessive glutamate concentrations alter the NMDA/GABA receptor activation ratio, leading to abnormal and toxic intracellular calcium concentrations and culminating in apoptotic death [17,18,19]. The glutamatergic excitotoxicity is the basis of neuro-inflammatory processes observed in neurodegenerative [9,20], ischemic, epileptic, traumatic and psychiatric diseases [15,21].

This evidence led to the rationale for the use of adamantanes, a family of molecules with a low affinity antagonism for the NMDA receptor [22]. The low affinity and rapid off-rate kinetic profile make adamantanes the best tolerated antiglutamatergic agents, considering the dangerous effects on learning and memory processes in case of excessive neuronal inhibition by a total glutamate inhibition [19]. Their prevalent adverse effects are headache, dizziness, hypertension, fatigue, constipation and nausea [23,24]. Adamantanes increase dopaminergic transmission [25,26,27], genetic expression of Glial cell-derived neuronal growth factor (GDNF) [28,29] and show many other neuroprotective effects [30,31,32]. All of this explains the plethora of potential clinical applications and research fields, from microglial neuroinflammation [33] to cerebral infarction [34], cerebral haemorrhage [35], traumatic brain injury [36], and degenerative disease [24,32,37,38,39,40,41]. Among the tested molecules, the best results in terms of safety and efficacy were recorded using amantadine in Parkinson’s disease [22,25,42,43] and memantine in Alzheimer’s disease and Lewy body dementia [44,45,46,47]. In particular, memantine is now widely used since its approval by the FDA for the treatment of moderate-severe Alzheimer’s disease, as monotherapy or combined with acetylcholinesterase inhibitors [48,49,50].

#### 3.1.2. Radiation-Induced Brain Toxicity

Radiation, like degenerative disease-related vascular dysfunctions, is an important cause of glutamate-induced excitatory stress. Several preclinical studies elucidated the cascade of events culminating in a rapid proliferation of dendritic spines and synapses, resulting in abnormal excitatory signals, synaptic loss and irreversible histological alterations [51]. In particular, radiation damages hippocampal synaptic structures and prefrontal-hippocampal cortex connections, critical for the construction of memory contents [52] and providers of neural progenitors [53,54]. Although neurons are classically considered to be radioresistant due to their post-mitotic state, radiation-induced synaptic damage was shown to be an early event [55]. The targets of radiation are neurons, considered as late-responding cells with a low α/β ratio [56,57], and stromal and vascular cells. Damage to the microvascular endothelium leads to microangiopathies with accelerated atherosclerosis, reduced neuronal and glial proliferation [58], and white matter hypotrophy [59,60,61] up to necrosis [62]. Reperfusion is also considered to cause brain damage due to massive ischemia, followed by a wide distribution of reactive oxygen species to which the brain is highly sensitive [63]. More recently, however, is the evidence on the role of astrocytes and microglia in the inflammatory pathway [64] and of specific brain areas [65] called “reservoirs of neural progenitors”. The latter are particularly sensitive to radiation [66] and are able to differentiate stem cells into neurons during the entire lifespan [67]. Common findings after irradiation on MRI are leukoencephalopathy, diffuse white matter change with possible moderate enhancement due to periventricular demyelination, edema and radionecrosis [68,69,70,71,72,73]. The radiation-induced effects are divided into early, late-early and late [74,75].

The first symptoms occur within hours to days after irradiation and are caused by alteration of the blood–brain barrier with vasogenic edema and damage to the white matter. Furthermore, the loss of radiosensitive stem cells occurs in this phase, partly caused by inflammation [76]. The most frequent symptoms are sleepiness, confusion, short-term memory and attention deficits, and fatigue [74,77]. The latter are reversible, and although established guidelines are lacking, steroids are often used for the prevention and treatment of brain edema [78,79]. Due to transient demyelination [80], delayed-early effects occur after a few weeks and up to six months and are initially dose-dependent and subsequently dose-independent [81]. The most common of these effects are neurapraxia [82], lethargic syndrome, mental confusion, impaired cognitive function, and fatigue [83,84,85]. Moreover, the incidence of these effects shows a peak at the end of radiotherapy (RT), with subsequent improvement in the following 6–8 weeks and with resolution within a further 4–6 weeks [86]. The late effects (recorded after 6–12 months) result from chronic neuroinflammation with persistent demyelination, reduced neurogenesis due to the glial differentiation shift, microvascular damage with ischemia and hyperglutamatergic toxic state [87,88]. Symptoms and signs of late, irreversible and progressive damage mainly concern cognitive functions and especially attention, memory and executive functions [68,89]. Less frequent but disabling are ataxic gait, urinary incontinence [90], apathy, and pyramidal and extrapyramidal syndrome [91].

#### 3.1.3. Cognitive Decline (CD)

Cognition is a predictor of quality of life [92] and survival [93]. It requires sensory, memory, visuospatial processing, concentration, attention, thought, behavior, personality and mood [94]. Dysfunction in one of these domains can significantly impair an individual’s communication and language skills, damaging their independence and professional and social functioning [95]. CD is common after therapeutic or prophylactic WBRT [91,96,97,98], particularly among adult subjects irradiated in the pediatric age [99,100] who often show impaired memorization of new content as well as worsened rapid information processing and attention retention [60,100]. Risk factors for the development of CD are age (<7 years, >60 years), large irradiation volume, high dose per fraction, chemotherapy, impaired pre-irradiation functional status [101,102,103], and vascular damage from hypertension and/or diabetes [86,104]. CD most commonly presents with memory loss, an impaired ability to plan activities, and behavioral changes. Radiation-induced CD is diagnosed in approximately 90% of irradiated patients and has intermediate severity in most subjects. However, CD can evolve into dementia in 2–5% of cases after irradiation with the most common fractionation protocols [89,90,105]. CD is worsened by intracranial tumor progression [91,106,107], antiepileptic drugs [108], chemotherapy [109], paraneoplastic syndromes [110] and corticosteroids. The latter has a well-known dose-dependent impact on mood and circadian rhythms [111] and treatments over six months may be associated with hippocampal hypotrophy with impaired memory function [112].

Several neuro-physiopathological studies focused on the loss of neural progenitors in the subgranular area of the hippocampus, the anatomical site of neural stem cells and thus on regenerative processes involved in the replacement of depleted neurons. In case of inflammation, the stem cells of the subgranular zone promote a maturative shift towards gliogenesis by reducing neurogenesis and cell population in the sites involved in the construction of memory contents [67,74,113,114]. Furthermore, interest also grew in the role of the extra-hippocampal regions, particularly the prefrontal cortex [115,116]. In fact, the radiation-induced acute and delayed effects on synaptic plasticity are proven even in “non-neurogenic” areas [116] with consequent functional damage to the entire neural “connectivity” [116].

#### 3.1.4. Radiation-Induced Cognitive Decline (RICD)

WBRT, formerly considered the standard of care for BMs [2,117], is now widely questioned due to the risk of CD. This negative impact of WBRT was clearly observed in clinical studies comparing the latter plus radiosurgery versus radiosurgery alone [97,118] and led to reduced use, especially in oligometastatic patients [119]. Furthermore, WBRT-induced CD was well documented in pediatric patient trials [120,121]. Moreover, prophylactic cranial irradiation (PCI) studies in adult subjects [122,123] showed acute CD, with relevant symptoms up to 2–4 weeks after completion of RT [96]. Finally, it should be noted that PCI doses (used for micrometastatic disease) are lower than those of therapeutic WBRT (for macroscopic metastases) and that acute RICD is not always followed by complete recovery [124]. Figure 1 summarizes the mechanisms of the radiation-induced cognitive deficit and the possible methods of prevention.

### 3.2. Main Evidence

#### 3.2.1. RTOG 0614 Study

The histopathological similarities between radiation-induced brain damage and vascular dementia [125,126,127] and the evidence of the post-irradiation NMDA-mediated hyperglutamatergic state led investigators to test the impact of anti-dementia drugs on RICD [128]. The first study testing an antiglutamatergic drug (memantine) to prevent RICD was the RTOG 0614 trial [129]. It was a randomized, double-blind, placebo-controlled study conducted in 143 centers across the United States and Canada, between March 2008 and July 2010, in WBRT-treated BM patients. The primary objective was to evaluate the impact of memantine-based prophylaxis on memory function six months after WBRT. The study was powered to detect a difference between the two arms of 0.87 in the Hopkins Verbal Learning Test-Revised (HVLT-R) score at 24 weeks. Secondary objectives were the impact of memantine on the executive, attentional and processing cognitive performances, and on survival outcomes. Inclusion criteria were: good Karnofsky performance status, no serious internal diseases, stable systemic disease for at least three months, non-impaired cognitive function upon the Mini-Mental State Examination, no allergy to memantine, no alcohol abuse, no chronic benzodiazepine intake, and no severe comorbidities. Cognitive function was assessed based on different domains using specific neuropsychological tests: the HVLT-R for memory and learning, the Controlled Oral Word Association test for verbal fluency, and the Trial Making Test (TMT) for processing speed and executive functions. Patients were randomized to receive WBRT plus placebo versus WBRT plus memantine. The WBRT dose was 37.5 Gy in 15 fractions, while the memantine intake started within three days of the first day of WBRT with progressive titration up to a dose of 20 mg daily (in two daily oral administrations) for a total of 24 weeks. Patients underwent general clinical, neurological and neuropsychological evaluation at baseline and at 8, 16, 24, and 52 weeks from treatment start. Five hundred and fifty-four patients were enrolled in the study and 508 were considered eligible.

The trial failed to achieve a statistically significant result regarding the primary endpoint, with the mean decline in the HVLT-R DR test in the memantine and placebo groups being 0 and 0.9 at 24 weeks, respectively (*p* = 0.059). The authors pointed out that only 149 patients were evaluable at 24 weeks, with a statistical power of only 35% to detect an absolute difference of 0.87. This large dropout rate was attributed to the high percentage of patients with tumor progression and death in the first months after WBRT. However, the trial achieved significant differences in secondary endpoints. Indeed, the time to CD, defined as the first decline on any neurocognitive test or a 2 SD drop from baseline for any test, was significantly improved in the memantine arm (HR 0.87, 95% CI, 0.62 to 0.99; *p* = 0.01) with a 21% lower chance of CD at 24 weeks (absolute rates: 53.8% vs. 64.9%). Moreover, the benefit in executive functions was observed starting from the eighth week. Furthermore, no differences in survival and toxicity rates were observed. In fact, memantine was well tolerated, as already observed in double-blind, placebo-controlled trials in dementia [130]. Finally, this trial was considered an example of a clinically but not statistically significant study [131].

#### 3.2.2. RTOG 0614 Trial-Related Studies

Based on the RTOG 0614 results, other studies evaluated antiglutamatergic drugs in brain-irradiated patients. One of these analyses tested DCE-MRI (dynamic contrast MRI) as a possible biomarker of WBRT-induced brain toxicity. Indeed, DCE-MRI was able to detect slight changes in blood–brain barrier permeability even in the early stages of Alzheimer’s disease [132,133,134] as well as in patients irradiated with high doses [135] or undergoing focused ultrasound [136] for glioma. Therefore, DCE-MRI was evaluated in detecting changes in vascular permeability in the six months after WBRT. The study included patients from the ROTG 0614 trial and was based on the evaluation of NAWM (normal appearance of white matter). The authors reported that even with WBRT doses (EQD2: 42.2 Gy, assuming α/β = 2), the vascular damage is detectable on the brain DCE-MRI and that changes in NAWM can predict RICD [137]. Furthermore, minor changes in NAWM were recorded in patients treated with prophylactic memantine [137], confirming the neuro- and vascular-protective role of antiglutamatergic drugs.

A subanalysis of the RTOG 0614 trial evaluated the correlation between health-related quality of life and cognitive function. Of the 447 patients included in the analysis, only 146 completed the questionnaires in week 24 [138]. No differences in quality of life were observed among the two arms, despite differences in objective cognitive function. The authors attributed this negative result to the preferential compilation of questionnaires by the best performing patients with a consequent overestimation of quality of life and an underestimation of CD data [31].

#### 3.2.3. Hippocampal-Avoidance: The RTOG 0933 Trial

Tsai et al., in a prospective study on 40 patients, reported a significant correlation between the RT dose delivered to the hippocampus and the incidence of RICD [139]. Therefore, in order to prevent RICD, WBRT techniques aimed at sparing the dentate gyrus of the hippocampus were developed. In particular, a highly conformed intensity-modulated technique was employed to bilaterally avoid the dentate gyrus of the hippocampus (hippocampal-avoiding-WBRT, HA-WBRT) [140,141]. Since the incidence of BM within 5 mm of the dentate gyrus is less than 5% [142], HA-WBRT was considered feasible also in terms of tumor control probability [142,143]. In the RTOG 0933 trial [144], 40 patients were treated with HA-WBRT and the results were compared to those of a historical WBRT group with similar clinical characteristics. The study demonstrated reduced CD by 4 months after RT and an improved quality of life after 6 months, confirming the neuroprotective effect of HA-WBRT.

#### 3.2.4. Pharmacological Prophylaxis plus Anatomical Sparing: The NRGCC001 Trial

Based on the RTOG 0614 and RTOG 0933 results, the NRGCC001 phase III trial was designed [145] to compare a standard WBRT arm versus a HA-WBRT arm with memantine prescribed in both arms. The primary endpoint was CD at four months, while secondary endpoints were overall survival, progression-free survival, toxicity and patients reporting quality of life. Brown et al. reported a significantly prolonged time to CD in the HA-WBRT arm (HR, 0.74; 95% CI, 0.58 to 0.95; *p* = 0.02). No significant differences were recorded at two months. However, significantly worse results in the WBRT arm were registered from the fourth month in terms of executive performances according to TMB-T (23.3% vs. 40.4%; *p*: 0.01), and at 6 months for learning and memory according to HVLTR TR (11.5% vs. 24.7%; *p*: 0.049; 16.4% vs. 33.3%; *p*: 0.02). No differences were recorded in terms of overall survival, progression-free survival, and toxicity. However, less fatigue (*p*: 0.04), less memory deficits (*p*: 0.01), less difficulty in speaking (*p*: 0.049), and less interference of neurological symptoms in daily activities (*p*: 0.008) were registered in the HA-WBRT arm. In the multivariate analysis, the only significant predictor of CD was age (>60 years), while RPA prognostic class, previous brain RT or surgery, and the volume of systemic disease did not show a significant impact. Despite the limitations of the trial (non-blinded, short follow-up: 7.9 months), these results led the authors to propose the combination of antiglutamatergic and hippocampal-sparing as a new standard of care in WBRT candidate patients, in good functional conditions, and with BMs at least 5 mm away from the hippocampal region. The results of the main trials on memantine combined with standard WBRT and HA-WBRT are shown in Table 1.

#### 3.2.5. Scientific Community Reactions

The proposal of the NRGCC001 trial researchers to redefine the therapeutic standard in patients with BM based on the results of their study has sparked a wide debate. In particular, many doubts have arisen about the interpretation of the results. [146,147]. Concerns about short-term results were: (1) at four months, the timing of the primary endpoint, HVLTR was completed by only 41% of patients; (2) at six months, the CD rate was high also in the memantine plus HA-WBRT arm; (3) the dropout rate at six months was very significant (experimental group: 48%; control group: 38%); (4) the time to decline of any neuropsychological test seems to be a non-specific endpoint to prove a significant clinical benefit.

Instead, the doubts about the long-term clinical outcomes of HA-WBRT were: (1) the impact of HA-WBRT on the long-term cognitive profile remains largely unknown (in the RTOG 0933 trial, the primary endpoint was assessed at four months); (2) a greater short-term tolerance after HA-WBRT may be associated with an increased risk of late microvascular complications; (3) in particular, a higher incidence of leukoencephalopathy was observed later after HA-WBRT compared to the traditional technique with opposite lateral fields [148].

Other reported issues were the following: (1) the possible imbalance in the bioprofile of the patients suggested by the separation (even if not significant) between the two survival curves; (2) the failure to evaluate possible bio-molecular parameters with prognostic impact [149,150] and the impact of immune checkpoint inhibitors (nivolumab, pembrolizumab) on cognitive function, potentially different from that of chemotherapy; (3) the failure to consider the cumulative volume of metastatic disease, potentially impacting cognitive function [151,152]; (4) the lack of stratification between BM at onset or in relapse/progression with consequently marked inhomogeneity of the patient population; (5) the failure to assess smoking and comorbidities (vascular diseases, diabetes) potentially related to neurological symptoms [104,153]; (6) the lack of consideration of systemic treatments which are potentially related to CD [123,154].

Based on these concerns, several authors defined it premature to consider memantine combined with anatomic sparing as a new therapeutic standard for BM. Furthermore, they considered more mature neurocognitive results and/or new studies considering potentially confounding factors and analyses of the patients’ reported outcome measures, as strongly needed [146,147,155]. However, the authors of the NRGCC001 study, regarding the comments on their trial, responded as follows [156]: (1) preventing RICD is very important for patients; (2) non-compliance and deaths did not compromise the NRGCC001 results, as the adherence rates in the two arms were similar; (3) the randomization process should have balanced the impact of the possible confounding variables on the primary endpoint (CD); (4) the long-term results, presented in a preliminary form, showed a persistent benefit on neurocognitive function [157,158]; (5) the doubts raised about the possible HA-induced leukoencephalopathy are only hypothetical and therefore should not influence clinical choices. Finally, the authors reiterated that the methodological quality of the NRGCC001 trial is sufficient for the definition of new standards of care, also according to the NCCN guidelines [159].

#### 3.2.6. State of Art

The international guidelines are contradictory in considering memantine as a new therapeutic standard in the BM setting. In fact, the NCCN guidelines included this indication [159] while the National Institute for Health and Care Excellence (NICE) guidelines are still awaiting further confirmation [160]. These conflicting opinions influenced the current clinical practice. In fact, an analysis of the SEER database was conducted, regarding the prescription of antiglutamatergic drugs for neurocognitive prophylaxis, on 6220 BM patients older than 65 years and treated with WBRT. The study included subjects treated between 2007 and 2016 and considered 2013 (publication date of the RTOG 0614 trial) as a “watershed”. Only 1.10% of patients received memantine between 2007 and 2013, while the rate rose, albeit slightly (5.14%), between 2013 and 2016 [161]. Subsequently, these figures further increased (9.36%) [161] but still suggest a widespread distrust of the role of antiglutamatergic therapy in reducing RICD.

In general, a greater interest in CD prophylaxis in patients undergoing WBRT would be conceivable, considering: (i) the improved prognosis potentially resulting from an earlier BM diagnosis and from the availability of more effective systemic treatments in several solid tumors [162]; (ii) the persistent use of WBRT in at least one-quarter of the BM patients [163]; (iii) the lack of availability in some centers, especially in less-resourced settings, of stereotactic RT or techniques capable of delivering HA-WBRT; (iv) the low cost (<1 dollar/day in most countries) and tolerability of memantine.

Moreover, even if only partially accepted by guidelines and in clinical practice, the aforementioned topics stimulated the design of several studies of antiglutamatergic prophylaxis in the context of RT of pediatric/young adult primary brain neoplasms (NCT03194906), of RT in the head and neck tumors (NCT03342443), of neuroprotection during breast cancer chemotherapy (NCT04033419), and in terms of comparison between memantine plus HA versus radiosurgery in patients with 5–15 metastases (NCT03550391).

Surprisingly enough, clinicians seem to prescribe memantine more often during prophylactic cranial irradiation (PCI) despite the lack of studies in this setting [164]. Furthermore, PCI, for extensive-stage small-cell lung cancer, is nowadays a controversial area after the publication of the results of a Japanese trial [165] questioning the survival benefit after PCI in subjects with extensive-stage small-cell lung cancer [166,167]. In particular, it is believed that PCI needs to be reconsidered, given the negative impact on cognition and, at the same time, the growing role of MRI surveillance and anti-check point agents [168,169,170,171,172,173].

### 3.3. New Scenarios

#### The Antitumor Effect of Memantine

Recent findings showed the important role of glutamate in the pathogenesis of glioblastoma (GBM). In fact, glutamate overexpression was found in the GBM models, both in the in vitro and in vivo studies [18], revealing the ability of these tumors to spontaneously produce concentrations up to four times the normal [16,174,175,176]. Moreover, the excess glutamate leads to the hyperactivation of N-methyl-D-aspartate receptors (NMDAR) with local excitotoxicity, neuroinflammatory cascade and cellular necrosis, typical of GBM [18,177]. Furthermore, NMDA stimulation promotes the growth and pro-invasive release of matrix metallo-proteinase-2 (MMP-2) from GBM cells [178]. Moreover, recent studies also suggest that GBM malignancy may depend on increased gliomagenesis due to the AKT pathway activation [179]. The discovery of the multiple pro-oncogenic effects of glutamate renewed the interest in antiglutamatergic agents, particularly NMDAR inhibitors [180]. In fact, memantine has shown the ability to: (i) inhibit the proliferation of GBM and medulloblastoma cell lines, even at the tumor penumbra level, populated by more invasive cellular subclones [179,181,182,183] and (ii) limit invasive motility through the reduction of pseudopodal protrusions [181]. Finally, in order to block the main oncogenetic pathways, prodrugs containing memantine were developed [184] and combinations of memantine and nitric oxide [185] or temozolomide were tested in the adjuvant setting [186].

## 4. Discussion

Neuro-oncological research is increasingly oriented towards strategies with an improved therapeutic index through higher local control and lower neurotoxicity rates. Furthermore, it was increasingly evident that cognitive function largely depends on extensive neuronal connectivity between the basal ganglia and the fronto-parietal cortex. This “neural network” is damaged by radiation-induced neuroinflammation, systemic therapies, surgery and local tumor progression through a hyperglutamatergic (with NMDAR hyperactivation) and excitotoxic state. The bio-molecular, histological and clinical similarities between vascular and radio-induced decline suggested a possible role of memantine, which is already used in the primary prevention of neurodegenerative vascular diseases, also in the prophylaxis of radiation-induced damage. In patients treated with WBRT, memantine was shown to delay CD time and partially limit its severity. Furthermore, memantine is well tolerated, even in the case of concomitant antiepileptics, steroids, antidepressants, hormone therapy and chemotherapy. Moreover, it is the only prophylaxis option in those countries where the available RT technology does not allow normal tissue sparing. The primary endpoint (CD) of the main trials was evaluated at six (RTOG0614) and four months (NRGCC001), being traditionally considered as a late effect. However, most CD occurs within the first two months after treatment. This “early” CD, recorded in several studies [96,187], was attributed to a general neurological deterioration from systemic therapies, WBRT, and above all, from “subclinical” tumor progression [129,131,188].

However, these interpretations are not entirely convincing. In fact, the microscopic disease is effectively controlled by radiation doses even lower than those delivered by WBRT, as demonstrated by PCI. More specifically, it is unclear why 25 Gy in 10 fractions (PCI) could be more effective than 30 Gy in 10 fractions (WBRT) in microscopic tumor control. Furthermore, it is not clear how a subclinical disease progression can compromise the neural network up to a “cognitive breakdown”. However, there is a lack of studies on the causes of early CD, historically attributed to white matter damage from vascular changes as well as a loss of radiosensitive stem cells from inflammatory mechanisms [76]. This depletion of neurogenic stem cells in the hippocampal dentate gyrus represented the rationale for the development of HA-WBRT. Nevertheless, HA-WBRT combined with memantine (in the experimental arm of the NRGCC001 trial) did not fully protect patients from relevant CD at 4 months. These data, together with the large loss of patients in the early follow-up period, require further reflection.

More precisely, the question could be: “Is there a link between early CD and severe patients’ loss in the early follow-up period?” A possible explanation seems to be provided by further recent neurophysiological evidence. In fact, according to the latter, the subgranular zone of the hippocampus and the subventricular zone of the lateral ventricles are not the only niches providing neural progenitors [189]. The first demonstrations of a large periventricular area expressing neurogenesis markers, also in adults, date back to 2000. This region also includes, in addition to the already known subgranular and subventricular zones, the limbic system and the hypothalamus [190,191]. Therefore, since these latter regions are also sites of neurogenesis, they should also be radiosensitive acute-responders, contrary to the identification of most brain tissue as a late-response tissue due to its low mitotic activity. Furthermore, the functional importance of these anatomical structures could help explain the severity of “early” neurological effects. In fact, the hypothalamus is responsible for the regulation of energy metabolism, reproduction, thermoregulation and circadian rhythms, and is involved in the processes of aging. Instead, the limbic system plays a key role in emotional reactions, behavioral responses, memory and smell. The existence of an early radio-induced “shock” on these neurogenic structures could explain the rapid onset signs and symptoms reported by patients undergoing WBRT as “profound fatigue”, the feeling of “mental fog”, a reduced “sense of taste and poor appetite”, and “not to feel oneself“ [192].

In other words, clinical frailty (weakness, fatigue) during and immediately after WBRT occurs simultaneously with the reduction in thyroid and sex hormone levels, recorded in the first months after WBRT [193]. Furthermore, an “acute illness syndrome” [194] was defined as anabolic hormone deficiency and poor sexual function. Therefore, together with the knowledge of CD after WBRT, awareness of neuroendocrine sequelae from the impaired hypothalamic-pituitary axis is growing [195]. Actually, the strong impact of irradiation on the hypothalamic-pituitary axis of pediatric subjects undergoing WBRT is well known but the possibility of a similar effect during WBRT for BM in adult subjects is recent [196]. In particular, there is a growing suspicion that acute hypothalamic dysfunction impairs the immune system’s ability to respond to infection and tumor progression. All this could explain, at least partially, the high patient dropout rates occurring shortly after WBRT.

## 5. Conclusions

Preserving cognitive functions is a priority for most BM patients [197,198]. Knowing the mechanisms of hyperglutamatergic neurotoxicity and the role of some hippocampal areas in CD led to testing both antiglutamatergic pharmacological prophylaxis and HA-WBRT techniques. These studies showed a relative reduction in CD four to six months after WBRT. However, failing to achieve statistical significance in the study by testing memantine alone (RTOG 0614) led to widespread skepticism about this drug in the WBRT setting, which may explain the lack of approval by the FDA and EMA of memantine in the prevention of radiation-induced CD. Moreover, interest grew in the reasons for the strong patient dropout in the first few months after WBRT and for early CD onset. In fact, the latter can only partially be explained by subclinical tumor progression. An emerging interpretation of the (not only) cognitive impairment during and immediately after WBRT is the dysfunction of the limbic and hypothalamic system with its immune and hormonal consequences. This new understanding of WBRT-induced toxicity may represent the basis for further innovative studies.

These studies should aim to: (i) evaluate in greater detail the cognitive effects and, more generally, the quality of life impairment during and immediately after WBRT; (ii) study the mechanisms producing these “early” effects; (iii) test in “clinical studies” advanced RT techniques based on both hippocampal-sparing and hypothalamic-pituitary-sparing, currently evaluated only in “planning studies” [196,199,200,201]; (iv) test new timings of antiglutamatergic drugs administration (e.g., start memantine a few weeks before WBRT), aimed at preventing not only late toxicity but also the acute effects; (v) compare stereotactic RT versus HA-WRT plus memantine based on data showing a reduced risk of CD in patients with 4–15 BMs treated with stereotactic RT compared to standard WBRT +/− memantine [202].

Finally, testing innovative RT techniques and pharmacological treatments to prevent CD would be particularly justified, given the more favorable prognosis compared to patients with BMs, in the settings of cerebral lymphomas [203] and low-grade gliomas [204,205].

## Figures and Tables

**Figure 1 cancers-14-02736-f001:**
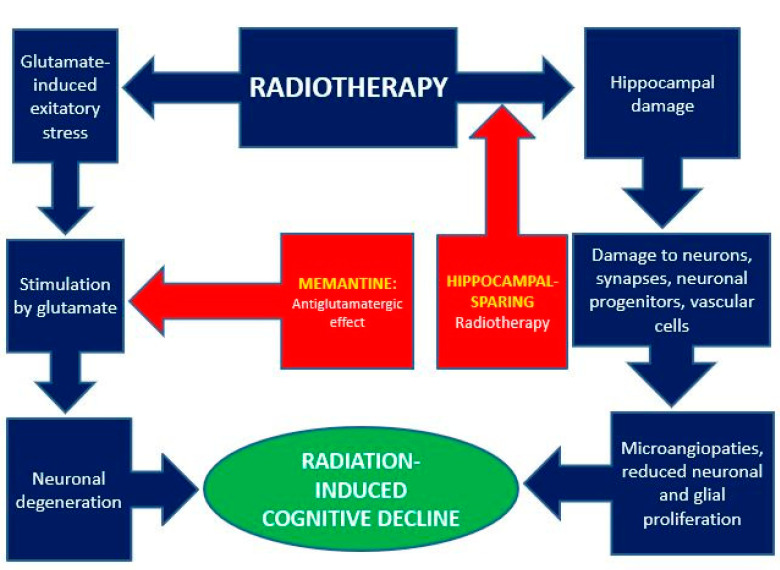
Mechanisms of radiation-induced cognitive impairment (blue) and possible ways of prevention (red).

**Table 1 cancers-14-02736-t001:** Results of the main trials on memantine combined with whole-brain radiotherapy and hippocampal-sparing whole-brain radiotherapy.

Authors/Year [Ref]	Trial	Study Design	Study Characteristic	Main Findings
Brown PD et al., 2013 [129]	RTOG 0614	Phase III, Random	Sample size and inclusion criteria: 508 adult patients with BM (only 149 evaluable at 24 weeks). Treatment: Patients were randomized to WBRT plus placebo versus WBRT plus memantine (20 mg/day for 24 weeks, starting within 3 days of WBRT start)	Similar grade 3–4 toxicity and study compliance in the two arms. Lower rate of decline in delayed recall (at 24 weeks) in the memantine arm but without reaching statistical significance (*p*: 0.059). Memantine arm: significantly longer time to CD (53.8% versus 64.9% at 24 weeks; HR: 0.78; 95% CI: 0.62–0.99, *p*: 0.01). better executive function at 24 weeks (*p*: 0.008 at 8 weeks; *p*: 0.0041 at 16 weeks), processing speed (*p*: 0.0137), and delayed recognition (*p*: 0.0149).
Laack NN et al., 2019 [138]	RTOG 0614 (subanalysis)	Phase III, Random	Subanalysis of the RTOG 0614 trial evaluating the correlation between health-related quality of life and cognitive function using FACT-Br and MOS-C	149 patients completed FACT-Br, MOS-C, and objective cognitive assessments at 24 weeks. Over time: worsening in all domains of objective cognitive function with no differences in FACT-Br and MOS-C between the 2 arms. improvement of emotional and functional well-being (FACT) with stability of the other FACT-Br domains. Conversely, declined MOS-C scores.
Tsai PF et al., 2015 [139]	RTOG 0933	Prospective	Sample size and inclusion criteria: 40 patients participated in an NCF assessment, including memory, executive function and psychomotor speed, before and after (4 months) HS-WBRT (assessments available in 24 patients). Treatment: therapeutic or prophylactic HS-WBRT. DVHs were generated for the left hippocampus, right hippocampus and hippocampal composite structure by calculating EQD2 (α/β: 2 Gy).	NCF scores are fairly stable before and after HS-WBRT in terms of hippocampus-dependent memory. EQD2 values < 12.60 Gy, <8.81 Gy, <7.45 Gy, and <5.83 Gy to 0%, 10%, 50%, and 80% volume of the hippocampal composite structure were significantly associated with preserved verbal memory. Specific dosimetric parameters of the left hippocampus impacted immediate recall of verbal memory (adjusted OR: 4.08; *p*: 0.042).
Brown PD et al., 2020 [145]	NRGCC001	Phase III, Random	Sample size and inclusion criteria: 518 adult patients with BM Treatment: Patients underwent HS-WBRT plus memantine versus WBRT plus memantine. Primary endpoint: time to CD (defined as a decline in at least one of the cognitive tests). Secondary endpoints: OS, intracranial PFS, toxicity and patient-reported symptom burden.	HS-WBRT arm: significantly lower CD risk (adjusted HR: 0.74; 95% CI: 0.58–0.95; *p*: 0.02), due to the lesser impairment of learning and memory at 6 months (11.5% versus 24.7% [*p*: 0.049] and 16.4% versus 33.3% [*p*: 0.02], respectively) and executive function at 4 months (23.3% versus 40.4%; *p*: 0.01). no differences in terms of OS, intracranial PFS and toxicity. at 6 months: less difficulty speaking (*p*: 0.049), less memory deficits (*p*: 0.01), and less fatigue (*p*: 0.04).

Legend: BM: brain metastases; CD: cognitive decline; DVH: dose-volume histograms; EQD2: biologically equivalent doses in fractions of 2 Gy; FACT-Br: Functional Assessment of Cancer Therapy-Brain module; HS-WBRT: hippocampal-sparing whole-brain radiotherapy; MOS-C: Medical Outcomes Scale-Cognitive Functioning Scale; NCF: neurocognitive function; OS: overall survival; PFS: progression-free survival; VMAT: volumetric modulated arch therapy; WBRT: whole-brain radiotherapy.

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
