# Peer review of "Memantine in the Prevention of Radiation-Induced Brain Damage: A Narrative Review"

_cancers, 2022, doi:10.3390/cancers14112736_

Round 1

Reviewer 1 Report

This narrative review was very interesting and important, but there were some points to revise before publishing.

1, The number of references are only 2 in Introduction section. Could you show us the references which you see when you wrote this section?

2, This article was narrative review, so the methods of search was not essential, I think. If you write this, could you write systematic review?

3, I think section 3. 1. 2 is similar to section 3. 1. 4. Could you put them together?

4, Could you add table for readers to understand easily?

5. Table 1 was too busy. Could you make easier to understand at a glance?

Reviewer 2 Report

Ref. cancers-1734752

The authors present a narrative review of the role of memantine in the prevention of radiation-induced brain damage in patients with neoplastic disease.

The review is well written and comprehensive.

 Minor points

Line 125: I would prefer to replace “…degenerative vascular disease…” with “Alzheimer’s disease”

Line 160: Do the authors mean “neurapraxia”

Line 169: Please change “…neuropathy..” to “syndrome”

Line 430: Cognitive function is the result of structural and functional integrity of the entire brain, including frontal, parietal, temporal and occipital lobes and their connections. Among the above, the connections between frontal and parietal cortex and basal ganglia/thalamus are very important for attention, concentration, procession speed, planning and executive function and for some aspects of memory.  I would suggest to rephrase the paragraph accordingly or simply change “…cognitive function is the result of….” to “…cognitive function largely depends on…”

Line 498: Is currently memantine FDA- or EMA-approved for use in such patients? If not, please state that clearly at this point.
